# Semantic Segmentation on Remotely Sensed Images Using an Enhanced Global Convolutional Network with Channel Attention and Domain Specific Transfer Learning

**Teerapong Panboonyuen [1,\*]**, **Kulsawasd Jitkajornwanich [2]**, **Siam Lawawirojwong [3]**, **Panu Srestasathiern [3]** and **Peerapon Vateekul [1,\*]**

[1] Chulalongkorn University Big Data Analytics and IoT Center (CUBIC), Department of Computer Engineering, Faculty of Engineering, Chulalongkorn University, Phayathai Rd, Pathumwan, Bangkok 10330, Thailand

[2] Data Science and Computational Intelligence (DSCI) Laboratory, Department of Computer Science, Faculty of Science, King Mongkut's Institute of Technology Ladkrabang, Chalongkrung Rd, Ladkrabang, Bangkok 10520, Thailand; kulsawasd.ji@kmitl.ac.th

[3] Geo-Informatics and Space Technology Development Agency (Public Organization), 120, The Government Complex, Chaeng Wattana Rd, Lak Si, Bangkok 10210, Thailand; siam@gistda.or.th (S.L.); panu@gistda.or.th (P.S.)

[\*] Correspondence: teerapong.panboonyuen@gmail.com (T.P.); peerapon.v@chula.ac.th (P.V.)

**Abstract:** In the remote sensing domain, it is crucial to complete semantic segmentation on the raster images, e.g., river, building, forest, etc., on raster images. A deep convolutional encoder–decoder (DCED) network is the state-of-the-art semantic segmentation method for remotely sensed images. However, the accuracy is still limited, since the network is not designed for remotely sensed images and the training data in this domain is deficient. In this paper, we aim to propose a novel CNN for semantic segmentation particularly for remote sensing corpora with three main contributions. First, we propose applying a recent CNN called a global convolutional network (GCN), since it can capture different resolutions by extracting multi-scale features from different stages of the network. Additionally, we further enhance the network by improving its backbone using larger numbers of layers, which is suitable for medium resolution remotely sensed images. Second, "channel attention" is presented in our network in order to select the most discriminative filters (features). Third, "domain-specific transfer learning" is introduced to alleviate the scarcity issue by utilizing other remotely sensed corpora with different resolutions as pre-trained data. The experiment was then conducted on two given datasets: (i) medium resolution data collected from Landsat-8 satellite and (ii) very high resolution data called the ISPRS Vaihingen Challenge Dataset. The results show that our networks outperformed DCED in terms of $F1$ for 17.48% and 2.49% on medium and very high resolution corpora, respectively.

**Keywords:** deep convolutional neural networks; multi-class segmentation; global convolutional network; channel attention; transfer learning; ISPRS Vaihingen; Landsat-8

## 1. Introduction

Semantic segmentation of earthly objects such as agriculture fields, forests, roads, and urban and water areas from remotely sensed images has been manipulated in many applications in various domains, e.g., urban planning, map updates, route optimization, and navigation [1–5], allowing us to better understand the domain's images and create important real-world applications.

A deep convolutional neural network (CNN) is a well-known method for automatic feature learning. It can mechanically learn features at different levels and abstractions from raw images by multiple hierarchical stacking convolution and pooling layers [4–14]. To accomplish such a challenging task, features at different levels are required. Specifically, abstract high-level features are more suitable for the recognition of confusing manmade objects, while the labeling of finely structured objects could benefit from detailed low-level features [1]. Therefore, different numbers of layers will affect the performance of deep learning models.

In the past few years, the modern CNNs have been extensively proposed including Global Convolutional Network (GCN) [15] in which the large kernel and effective receptive field play an important role in performing classification and localization tasks simultaneously. The GCN is proposed to address the classification and localization issues for semantic segmentation and to suggest a residual-based boundary refinement for further refining object boundaries. However, this type of architecture ignores the global context such as weights of the features in each stage. Furthermore, most methods of this type are just summed up the features of adjacent stages without considering their diverse representations. This leads to some inconsistent results that suffer from accuracy performance. The primary challenge of this remote sensing task is a lack of training data. This, in fact, has become a motivation of this work.

In this paper, we present a novel global convolutional network for segmenting multi-objects from aerial and satellite images. To this end, it is focused on three aspects: (i) varying backbones using ResNet50, ResNet101, and ResNet152, (ii) applying a "channel attention block" [16,17] to assign weights for feature maps in each stage of the backbone architecture, and (iii) employing "domain-specific transfer learning" [18–20] to relieve scarcity. Experiments were conducted using satellite imagery (from the Landsat-8 satellite), which was provided by a government organization in Thailand, and using well-known aerial imagery from the ISPRS Vaihingen Challenge corpus [21], which is publicly available. The results showed that our method outperforms the baseline including deep convolutional encoder–decoder (DCED) in terms of $F1$ and by mean of class-wise intersection over union (*mean IoU*).

The remainder of this paper is arranged as follows. The related work is discussed in Section 2. Section 3 describes our proposed methodology. Experimental datasets and evaluations are described in Section 4. Experimental results and discussions are presented in Section 5. Finally, we conclude our work and discuss future work in Section 6.

## 2. Related Work

Deep learning has been successfully applied for remotely sensed data analysis, notably land cover mapping on urban areas [1–3], and has increasingly become a promising tool for accelerating the image recognition process with high accuracy [4–14,22–30]. It is a fast-growing field, and new architectures appear every few days. This section is divided into three subsections: we discuss deep learning concepts for semantic segmentation, a set of multi-objects segmentation techniques using modern deep learning architectures, and modern techniques of deep learning.

### 2.1. Deep Learning Concepts for Semantic Segmentation

Semantic segmentation algorithms are often formulated to solve structured pixel-wise labeling problems based on a deep CNN. Noh et al. [13] proposed a novel semantic segmentation technique utilizing a deconvolutional neural network (DCNN) and the top layer from the DCNN adopted from

VGG16 [4,8]. The DCNN structure is composed of upsampling layers and deconvolution layers, describing pixel-wise class labels and predicting segmentation masks, respectively. Their proposed deep learning methods yield high performance in PASCAL VOC 2012 corpus, with the 72.5% accuracy in the best-case scenario (the highest accuracy—as of the time of the writing of this paper—compared to other methods that were trained without requiring additional or external data). Long et al. [12] proposed adapted contemporary classification networks incorporating Alex, VGG, and GoogLe networks into a fully CNN. In this method, some of the pooling layers were skipped: Layer 3 (FCN-8s), Layer 4 (FCN-16s), and Layer 5 (FCN-32s). The skip architecture reduces the potential over-fitting problem and has shown improvements in performance, ranging from 20% to 62.2% in the experiments tested on PASCAL VOC 2012 data. Ronneberger et al. [14] proposed U-Net, a DCNN for biomedical image segmentation. The architecture consists of a contracting path and a symmetric expanding path that captures context and consequently enables precise localization. The proposed network claimed to be capable of learning despite the limited number of training images and performed better than the prior best method (a sliding-window DCNN) on the ISBI challenge for segmentation of neuronal structures in electron microscopic stacks. Vijay Badrinarayanan [31–33] proposed a deep convolutional encoder–decoder network (DCED), called "SegNet," that consists of two main networks, encoder and decoder, and some outer layers. The two outer layers of the decoder network are responsible for feature extraction, the results of which are transmitted to the layer adjacent to the last layer of the decoder network. This layer is responsible for pixel-wise classification (determining which pixel belongs to which class). There is no fully connected layer in between feature extraction layers. In the upsampling layer of the decoder, pool indices from the encoder are distributed to the decoder, where the kernel will be trained in each epoch (the training round) at the convolution layer. In the last layer (classification), softmax was used as a classifier for pixel-wise classification. The DCED is one of the deep learning models that exceeds the state of the art on many remote sensing corpus.

In this work, the DCED method was selected as our baseline since it is the most popular architecture used in various networks for semantic segmentation.

## 2.2. Modern Deep Learning Architectures For Semantic Segmentation

Recently, many approaches based on the DCED have achieved high performance on different benchmarks [16,31–33]. However, most of them still suffer from accuracy performance issues. Therefore, many works of modern deep learning architectures have been proposed, such as instance-aware semantic segmentation [34], which is slightly different from semantic segmentation. Instead of labeling all pixels, it focuses on the target objects and labels only pixels of those objects. FCIS [28] is based on techniques based on fully convolutional networks (FCNs). The mask R-CNN [9] was built around the FCN and is incorporated with a proposed joint formulation. Peng [15] presented the concept of large kernel matters to improve semantic segmentation with a global convolutional network (GCN) as shown in Figure 1. They proposed a GCN to address both the classification and localization issues for semantic segmentation. Large separable kernels were used to expand the receptive field, and a boundary refinement block was added to further improve localization performance near the boundaries. From the Cityscapes challenge, the GCN outperforms methods of all previous publications (all modern deep learning baselines) and has become the new state of the art. Therefore, the GCN was selected as our proposed method and as the main model of our work.

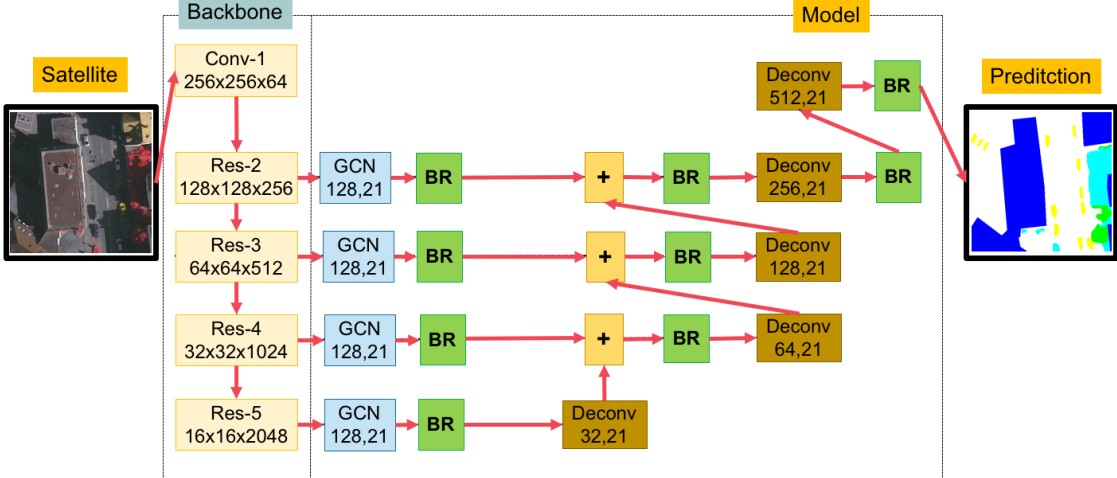

**Figure 1.** An overview of the original global convolutional network (GCN) and boundary refinement (BR) [15].

### 2.3. Modern Techniques of Deep Learning

Modern techniques of deep learning are important for the accuracy of a CNN. The most popular modern ideas used for semantic segmentation tasks, such as global context, the attention module, and semantic boundary detection, have been used for boosting accuracy.

Global context [16] is a modern method that has proven the effectiveness of global average pooling in the semantic segmentation task. For example, PSPNet [30] and Deeplab v3 [5] respectively extend it to spatial pyramid pooling [30] and atrous spatial pyramid pooling [5], resulting in great performance at different benchmarks. However, to take advantage of the pyramid pooling module sufficiently, these two methods adopt the base feature network to downsample with atrous convolution eight times [5], which is time-consuming and memory-intensive.

Attention Module [16]: Attention is helpful to focus on what we want. Recently, the attention module has increasingly become a powerful tool for deep neural networks [16,17]. The method in [16,17] pays attention to different scale information. In this work, we utilize a channel attention block to select features, similar to learning a discriminative feature network [16].

Refinement Residual Block [16]: The feature maps of each stage in the feature network all go through the refinement residual block. For our work, we use the boundary refinement block (BR) to be a concept of "refinement residual block" from [15]. The first component of the block is a $1 \times 1$ convolution layer. We use it to unify the number of channels to 21. Meanwhile, it can combine the information across all channels. Then the following is a basic residual block [7], which can refine the feature map. Furthermore, this block can strengthen the recognition ability of each stage, inspired from the architecture of ResNet.

## 3. The Proposed Method

In this section, the details of our proposed network are explained (shown in Figure 2). The network is based on the GCN with three aspects of improvements: (i) the modification of backbone architecture (shown in P1 in Figure 2), (ii) applying the channel attention block (shown in P2 in Figure 2), and (iii) using the concept of domain-specific transfer learning (shown in P3 in Figure 2).

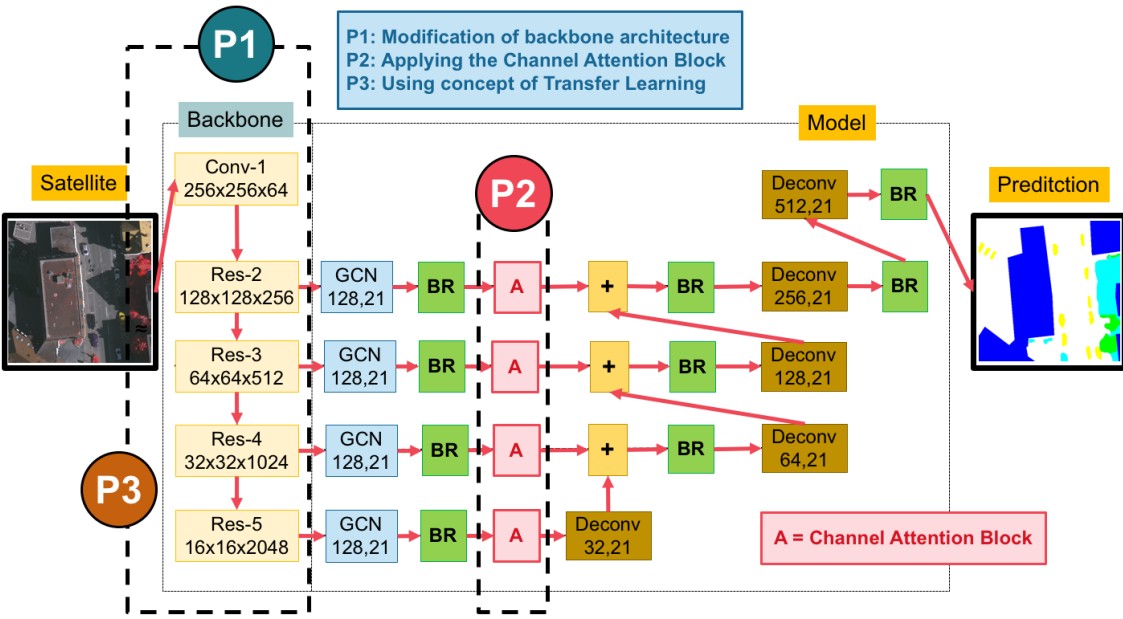

**Figure 2.** An overview of our proposed network.

## 3.1. Data Preprocessing

In this paper, there are two benchmark corpuses, including (i) the ISPRS Vaihingen Challenge corpus and (ii) the Landsat-8 dataset. They are comprised of very high and medium resolution images, respectively. More details of the datasets will be explained in Sections 4.1 and 4.2. Before a discussion of the model, it is worth explaining our data preprocessing procedure, since it is required when working with neural network and deep learning models. Thus, the mean subtraction is executed.

In addition, data augmentation is often required on more complex object recognition tasks. Therefore, a random horizontal flip is generated to increase the training data. For the ISPRS corpus, all images are standardized and cropped into $512 \times 512$ pixels with a resolution of 9 cm$^2$/pixel. For the Landsat-8 corpus, each image is also flipped horizontally and scaled to $512 \times 512$ with a resolution of 30 m$^2$/pixel from the original images (16,800 $\times$ 15,800 pixels).

## 3.2. A Global Convolutional Network (GCN) with Variations of Backbones

GCN [15] as shown in Figure 1 is a modern architecture that surpasses the drawbacks of a traditional semantic segmentation network, such as deep convolutional encoder–decoder (DCED) networks. A traditional network usually cascades convolutional layers in order to generate sophisticated features; they can be considered as local features that are specialized for a specific task. However, it is not necessary to employ only specialized features; the general features are also important. Thus, a GCN overcomes this issue by introducing a multi-level architecture, where each level aims to capture a different resolution of features, so both local and global features are considered in the model.

As shown in Figure 1, there are two main blocks in the GCN: a localization block and a classification block. From the localization view in the left block, the structure is a stack of classical fully convolutional layers called "levels." Each level aims to construct features with different resolutions. From the classification view, there are two modules: the GCN and the boundary refinement (BR). For the GCN module, the kernel size of the convolutional structure should be as large as possible, which is motivated by the densely connected structure of the classification models. If the kernel size increases to the spatial size of the feature map (named the global convolution), the network will share the same benefits with the pure classification models. The BR module is added to further improve localization performance near the boundaries.

Although the GCN architecture has shown promising prediction performance, it is still possible to further improve by varying backbones using ResNet [7] with different numbers of layers as ResNet50, ResNet101, and ResNet152, as shown in Figure 3. Additionally, the GCN is suggested to work on a large kernel size. In this paper, we set the large kernel size as 9 (this previous work [15]).

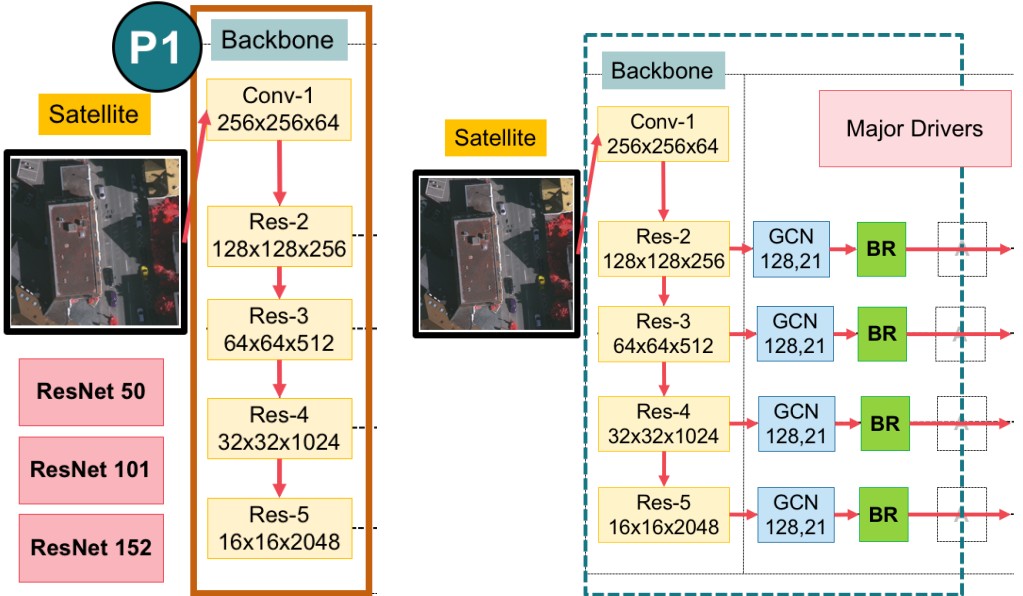

**Figure 3.** An overview of the whole backbone pipeline in (**left**) the main backbone with varying by ResNet50, ResNet101, and ResNet152; (**right**) the major drivers of our main classification network (composed of a global convolutional network (GCN) and a boundary refinement (BR) block [15]).

### 3.3. The Channel Attention Block

Attention mechanisms [16,17] in neural networks are very loosely based on the visual attention mechanism found in humans and equips a neural network with the ability to focus on a subset of its inputs (or features): it selects specific inputs. Human visual attention is well-studied, and while there are different models, all of them essentially come down to being able to focus on a certain region of an image with a very high resolution, perceiving the surrounding image in a medium resolution, and then adjusting the focal point over time.

To apply this atttentional layer to our network, the channel attention block is shown in Block A in Figure 2 and its detailed architecture is shown in Figure 4. It is designed to change the weights of the remote sensing features on each stage (level), so that the weights are assigned more values on important features adaptively.

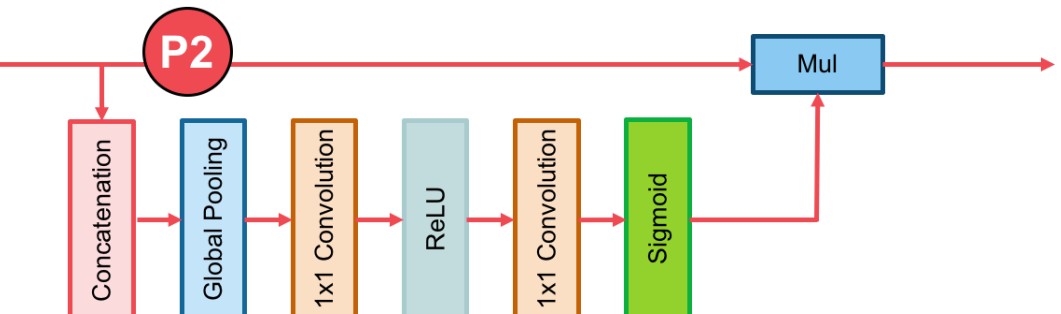

**Figure 4.** Components of the channel attention block. The red lines represent the downsample operators, respectively. The red line cannot change the size of feature maps. It is only a path for information passing.

In the proposed architecture, a convolution operator gives the probability of each class at each pixel. In Equation (1), the final score is summed over all channels of the feature maps.

$$y_k = F(x; w) = \sum_{i=1, j=1}^{D} w_{i,j} x_{i,j} \tag{1}$$

where $x$ is the output feature of network. $w$ represents the convolution's kernel, and $k \in$ 1, 2, 3, 4, 5, 6, 7,..., $K$. The number of channels is represented by $K$, and $D$ is the set of pixel positions.

$$\delta_i(y_k) = \frac{exp(y_k)}{\sum_{j=1}^{k} exp(y_j)} \tag{2}$$

where $\delta$ is the prediction probability. $y$ is the output of the network. As shown in Equations (1) and (2), the final predicted label is the category with the highest probability. Therefore, we suppose that the prediction result is $y_0$ of a certain patch, while its true label is $y_1$. Therefore, we can introduce a parameter $\alpha$ to change the highest probability value from $y_0$ to $y_1$, as Equation (3) shows.

$$\bar{y} = \alpha y = \begin{bmatrix} \alpha_1 \\ . \\ . \\ . \\ \alpha_k \end{bmatrix} \cdot \begin{bmatrix} y_1 \\ . \\ . \\ . \\ y_k \end{bmatrix} = \begin{bmatrix} \alpha_1 w_1 \\ . \\ . \\ . \\ \alpha_k w_k \end{bmatrix} \times \begin{bmatrix} x_1 \\ . \\ . \\ . \\ x_k \end{bmatrix} \tag{3}$$

where $\bar{y}$ is the new prediction of the network, and $\alpha = Sigmoid(x; w)$.

Based on the above formulation of the Channel Attention Block, we can explore its practical significance. In Equation (1), it implicitly indicates that the weights of different channels are equal. However, the features in different stages have different degrees of discrimination, which results in different consistency of prediction. Consequently, in Equation (3), the $\alpha$ value applies the feature maps $x$, which represents the feature selection with the channel attention block.

### 3.4. Domain-Specific Transfer Learning

The overall idea of transfer learning is to use knowledge learned from tasks for which many labeled data are usable in settings where only little-labeled data are available. Creating labeled data is expensive, so optimally leveraging an existing dataset is key. Certain low-level features, such as edges, shapes, corners, and intensity, can be shared across tasks, and new high-level features specific to the target problem can be learned [18]. Additionally, knowledge from an existing task acts as an additional input when learning a new target task.

Although the deep learning approach often performs promising prediction performance, it requires a large amount of training data. Since it is difficult to obtain annotated satellite images, the performance in prior works has been limited.

Fortunately, there is a recent concept called domain-specific transfer learning [18–20] that allows one to reuse the weights obtained from other domains' inputs. It is currently very popular in the field of deep learning because it enables one to train deep neural networks with comparatively insufficient data. This is very useful since most real-world problems typically do not have millions of labeled data points to train such complex models.

In terms of inadequacy, we propose an effective transfer deep neural network to perform knowledge transfer between a very high resolution (VHR) corpus and a medium resolution (MR) corpus. It is shown in Figure 5.

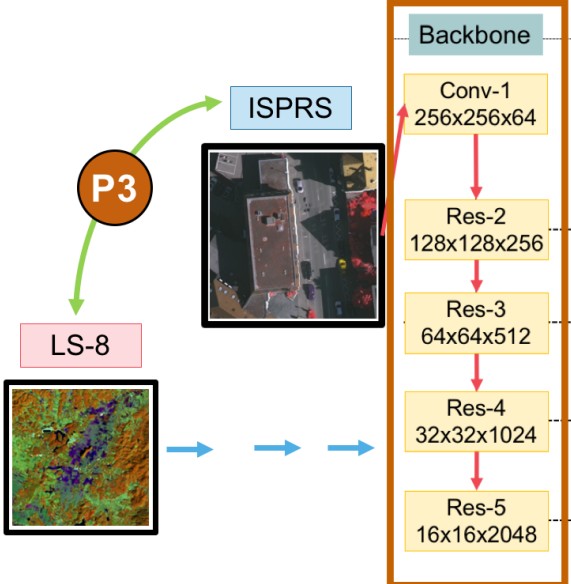

**Figure 5.** The domain-specific transfer learning strategy reuses pre-trained weights of models between two datasets—very high (ISPRS) and medium (Landsat-8; LS-8) resolution images.

## 4. Experimental Datasets and Evaluation

In our experiments, two types of datasets were used: (i) medium resolution imagery (satellite images; Landsat-8 dataset) made by the government organization in Thailand, named GISTDA (Geo-Informatics and Space Technology Development Agency (Public Organization)), and (ii) very high resolution imagery (aerial images; ISPRS Vaihingen dataset). All experiments were evaluated based on major metrics, such as *average accuracy*, *F*1 score, and *mean IoU* score.

### 4.1. Landsat-8 Dataset

Landsat-8 is an American earth observation satellite and it collects and archive medium resolution (30-m spatial resolution) multispectral image data affording seasonal coverage of the global landmasses for a period of no less than 5 years. Landsat-8 [35] images consist of nine spectral bands with a spatial resolution of 30 m for Bands 1–7 and 9. The ultra blue Band 1 is useful for coastal and aerosol studies. Band 9 is useful for cirrus cloud detection. The resolution for Band 8 (panchromatic) is 15 m. Thermal Bands 10 and 11 are useful in providing more accurate surface temperatures and are collected at 100 m. The approximate scene size is 170 km north–south by 183 km east–west (106 mi by 114 mi). Since Landsat-8 data includes additional bands, the combinations used to create RGB composites differ from Landsat 7 and Landsat 5. For instance, Bands 4, 3, and 2 are used to create a color infrared (CIR) image using Landsat 7 or Landsat 5. To create a CIR composite using Landsat 8 data, Bands 5, 4, and 3 are used.

In this type of data, the satellite images are from Nan, a province in Thailand. The dataset is obtained from Landsat-8 satellite consisting of 1012 satellite images as shown by some samples in Figure 6.

This corpus is comprised of a large, diverse set of medium resolution (16,800 × 15,800) pixels, where 1012 of these images have high quality pixel-level labels of five classes: agriculture, forest, miscellaneous, urban, and water. The 1012 images were split into 800 training and 112 validation images with publicly available annotation, as well as 100 test images with annotations withheld, and comparison to other methods were performed via a dedicated evaluation server. For quantitative evaluation, mean of class-wise intersection over union (*mean IoU*) and *F*1 score are used.

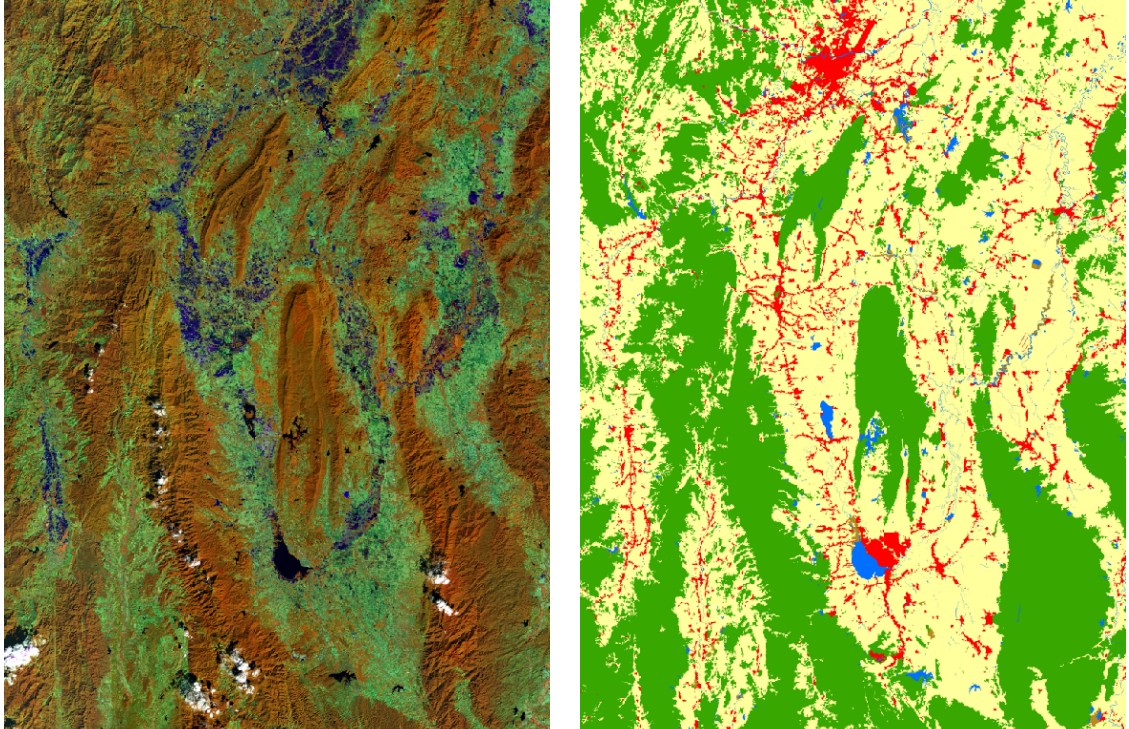

**Figure 6.** Sample satellite images from Nan, a province in Thailand (**left**), and corresponding ground truth (**right**). The label of medium resolution dataset includes five categories: agriculture (yellow), forest (green), miscellaneous (brown), urban (red), and water (blue).

### 4.2. ISPRS Vaihingen Dataset

One of the major challenges in remote sensing is the automated extraction of urban objects from data acquired by airborne sensors. The Semantic Labeling Contest provides two state-of-the-art airborne image corpora. The Vaihingen corpus shows a relatively small village with many detached buildings and small multi-story buildings, and the Potsdam corpus shows a typical historic city with large building blocks, narrow streets, and dense settlement structure. In our experiments, the Vaihingen corpus was selected and used.

The ISPRS 2D Semantic labeling challenge in Vaihingen [21] (Figures 7 and 8) was used as our benchmark dataset. It consists of three spectral bands (i.e., red, green, and near-infrared bands), the corresponding DSM (digital surface model) and the NDSM (normalized digital surface model) data. Overall, there are 33 images of about 2500 × 2000 pixels at a ground sampling distance (GSD) of about 9 cm in the image data. Among them, the ground truth of only 16 images are available, and those of the remaining 17 images are withheld by the challenge organizer for the online test. For offline validation, we randomly split the 16 images with ground truth available into a training set of 10 images and a validation set of 6 images. For this work, DSM and NDSM data in all experiments on this dataset were not used. Following other methods, four tiles (Image Numbers 5, 7, 23, and 30) were removed from the training set as the validation set. Experimental results are reported on the validation set if not specified.

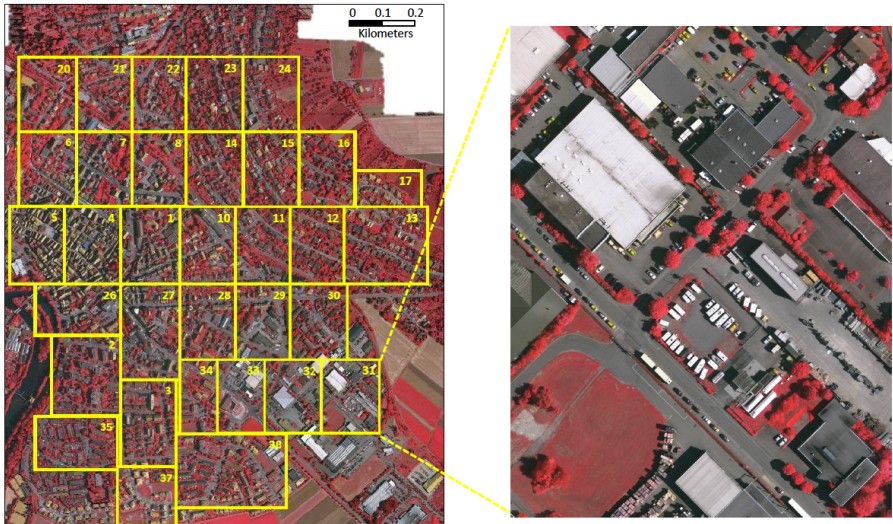

**Figure 7.** Overview of the ISPRS 2D Vaihingen Labeling corpus. There are 33 tiles. Numbers in the figure refer to the individual tile flag.

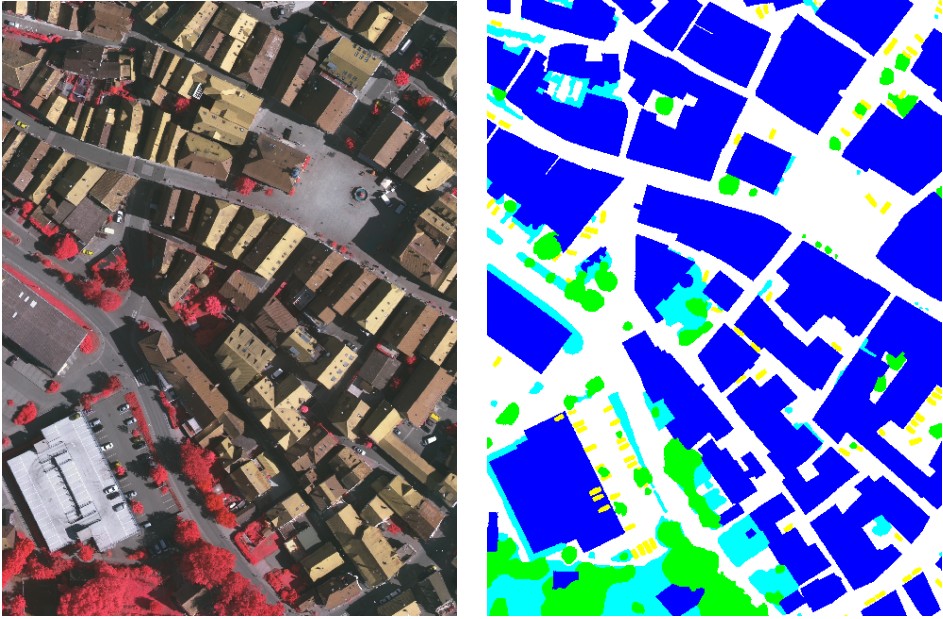

**Figure 8.** The sample input tile from Figure 7 (**left**) and corresponding ground truth (**right**). The label of the Vaihingen Challenge includes six categories: impervious surface (imp surf, white), building (blue), low vegetation (low veg, cyan), tree (green), car (yellow), and clutter/background (red).

### 4.3. Evaluation

The multi-class classification task can be considered as multi-segmentation, where class pixels are positives and the remaining non-spotlight pixels are negatives. Let $TP$ denote the number of true positives, $TN$ denote the number of true negatives, $FP$ denote the number of false positives, and $FN$ denote the number of false negatives.

*Precision*, *recall*, *F*1, and *mean IoU* are shown in Equations (4)–(8). Precision is the percentage of correctly classified main pixels among all predicted pixels by the classifier. Recall is the percentage of correctly classified main pixels among all actual main pixels. *F*1 is a combination of *precision* and *recall*.

To evaluate the performance of different deep models, we will discuss the above two major metrics (*F*1), the mean of class-wise intersection over union (*mean IoU*)) on each category, and the mean value of metrics to assess the average performance.

$$Accuracy = \frac{TP + TN}{TP + FP + FN + TN} \tag{4}$$

$$Precision = \frac{TP}{TP + FP} \tag{5}$$

$$Recall = \frac{TP}{TP + FN} \tag{6}$$

$$F1 = \frac{2 \times Precision \times Recall}{Precison + Recall} \tag{7}$$

$$Mean\ IoU = \frac{TP}{TP + FP + FN}. \tag{8}$$

## 5. Experimental Results and Discussion

The implementation is based on a deep learning framework, called "Tensorflow-Slim" [36], which is extended from Tensorflow. All experiments were conducted on servers with an Intel® Xeon® Processor E5-2660 v3 (25M Cache, 2.60 GHz), 32 GB of memory (RAM), an Nvidia GeForce GTX 1070 (8 GB), an Nvidia GeForce GTX 1080 (8 GB), and an Nvidia GeForce GTX 1080 Ti (11 GB). Instead of using the whole image (1500 × 1500 pixels) to train the network, we randomly cropped all images to be 512 × 512 as inputs of each epoch.

For training, the Adam optimizer [11] was chosen with an initial learning rate of 0.004 and the weight decay of 0.00001. Batch normalization [10] is used before each convolutional layer in our implementation to ease the training and make it be able to concatenate feature maps from different layers. To avoid overfitting, common data augmentations are used as details in Section 3.1. For measurements, we use the mean pixel intersection-over-union (*mean IoU*) and the *F1* score as the metric.

Inspired by [16,27,37], we use the "poly" learning rate policy where the learning rate is multiplied by Equation (9) with a power of 0.9 and an initial learning rate as $4 \times 10^{-3}$. The learning rate is scheduled by multiplying the initial as seen in Equation (9).

$$learning\ rate = (1 - \frac{epoch}{MaxEpoch})^{0.9}. \tag{9}$$

All models are trained for 50 epochs with a mini-batch size of 4, and each batch contains the cropped images that are randomly selected from training patches. These patches are resized to 521 × 521 pixels. The statistics of BN is updated on the whole mini-batch.

This section illustrates the details of our experiments. The proposed deep learning network is based on the GCN with three improvements: (i) varying the backbones using ResNet, (ii) channel attention and global average pooling, and (iii) domain-specific transfer learning. From all proposed strategies, there are six acronyms of strategies as shown in Table 1.

**Table 1.** Abbreviations on our proposed deep learning methods.

| Abbreviation | Description |
| --- | --- |
| A | Channel Attention Block |
| GCN | Global Convolutional Network |
| GCN50 | Global Convolutional Network with ResNet50 |
| GCN101 | Global Convolutional Network with ResNet101 |
| GCN152 | Global Convolutional Network with ResNet52 |
| TL | Domain-Specific Transfer Learning |

For the experimental setup, there were three experiments on two remotely sensed datasets: the Landsat-8 dataset and the ISPRS Vaihingen Challenge dataset (details in Sections 4.1 and 4.2). The experiments aimed to illustrate that each proposed strategy can improve the performance. First, the GCN152 method was compared to the GCN50 method and the GCN101 method for the varying backbones using ResNet with different numbers of layers on the GCN network strategy. Second, the GCN152-A method was compared to the GCN152 method for the channel attention strategy. Third, the full proposed technique GCN152-TL-A method was compared to existing methods for the concept of domain-specific transfer learning.

### 5.1. Results of the Landsat-8 Corpus with Discussion

An experiment was conducted on the Landsat-8 corpus, and the result is shown in Tables 2 and 3 by comparing between baseline and variations of the proposed techniques. It is shown that our network with all strategies, GCN152-TL-A, outperforms other methods. More details will be discussed to show that each of the proposed techniques can improve accuracy. Only in this experiment is there a state-of-the-art baseline, including a deep convolutional encoder–decoder (DCED) [31–33].

**Table 2.** Results of the testing data of the Landsat-8 corpus between baseline and five variations of our proposed techniques in terms of *precision*, *recall*, *F1*, and *mean IoU*.

|  | Pretrained | Backbone | Model | *Precision* | *Recall* | *F1* | *mean IoU* |
|---|---|---|---|---|---|---|---|
| **Baseline** | - | - | DCED [31–33] | 0.6137 | 0.7209 | 0.6495 | 0.5384 |
| **Proposed Method** | - | Res50 | GCN [15] | 0.6678 | 0.7333 | 0.6847 | 0.5734 |
| | - | **Res101** | GCN | 0.6899 | 0.8031 | 0.7290 | 0.6154 |
| | - | **Res152** | GCN | 0.7115 | 0.8131 | 0.7563 | 0.6364 |
| | - | **Res152** | **GCN-A** | 0.7997 | 0.7937 | 0.7897 | 0.6726 |
| | TL | **Res152** | **GCN-A** | **0.8293** | **0.8476** | **0.8275** | **0.7178** |

**Table 3.** Results of the testing data of Landsat-8 corpus between each class with our proposed techniques in terms of *average accuracy*.

|  | Model | Agriculture | Forest | Misc | Urban | Water |
|---|---|---|---|---|---|---|
| **Baseline** | DCED [31–33] | 0.9616 | 0.7472 | 0.0976 | 0.7878 | 0.4742 |
| **Proposed Method** | GCN50 [15] | 0.9407 | 0.8258 | 0.1470 | **0.8828** | 0.5426 |
| | GCN101 | 0.9677 | 0.8806 | 0.2561 | 0.7971 | 0.5480 |
| | GCN152 | 0.9780 | 0.8444 | 0.4256 | 0.7158 | 0.5937 |
| | **GCN152-A** | 0.9502 | **0.9118** | 0.6689 | 0.8675 | 0.6001 |
| | **GCN152-TL-A** | **0.9781** | 0.8472 | **0.8732** | 0.7988 | **0.6493** |

5.1.1. The Effect of an Enhanced GCN on the Landsat-8 Corpus

Our first strategy aims to increase an *F*1 and *mean IoU* score of the network by varying backbones using ResNet 50, ResNet 101, and ResNet 152 rather than the traditional one, the DCED method. From Tables 2 and 3, the *F*1 of GCN152 (0.7563) outperforms that of GCN50 (0.6847), GCN101 (0.7290), and the baseline method, DCED (0.6495); this yields a higher *F*1 at 2.74%, 3.52%, and 4.43%, respectively. The *mean IoU* of GCN152 (0.6364) outperforms that of GCN50 (0.5734), GCN101 (0.6154), and the baseline method, DCED (0.5384); this yields a higher *mean IoU* at 2.10%, 3.50%, and 4.20%, consecutively. The main reason is due to higher precision, but a slightly lower recall. This can imply that enhanced GCN is more significantly efficient than the DCED method (baseline) for this medium resolution corpus and ResNet with a large number of layers is more robust than the small number of layers.

When comparing the results between the original GCN method and the enhanced GCN methods on the Landsat-8 corpus (Table 2), it is clearly shown that a GCN with a larger layer of backbone can improve network performance in terms of *F*1 and *mean IoU*.

5.1.2. The Effect of Using Channel Attention on the Landsat-8 Corpus

Our second mechanism focused on applying the channel attention block (details in Section 3.4) to change the weights of the features on each stage to enhance consistency. In Tables 2 and 3, the *F1* of GCN152-A (0.7897) is greater than that of GCN152 (0.7563); this yields a higher *F1* score at 3.34%. The *mean IoU* of GCN152-A (0.6726) is superior to that of GCN152 (0.6364); this yields a higher *mean IoU* score at 3.62%. The result (Figures 9e and 10e) shows that can make the network to obtain discriminative features stage-wise to make the prediction intra-class consistent. This is based on the consideration that we re-weighted all feature maps of each layer.

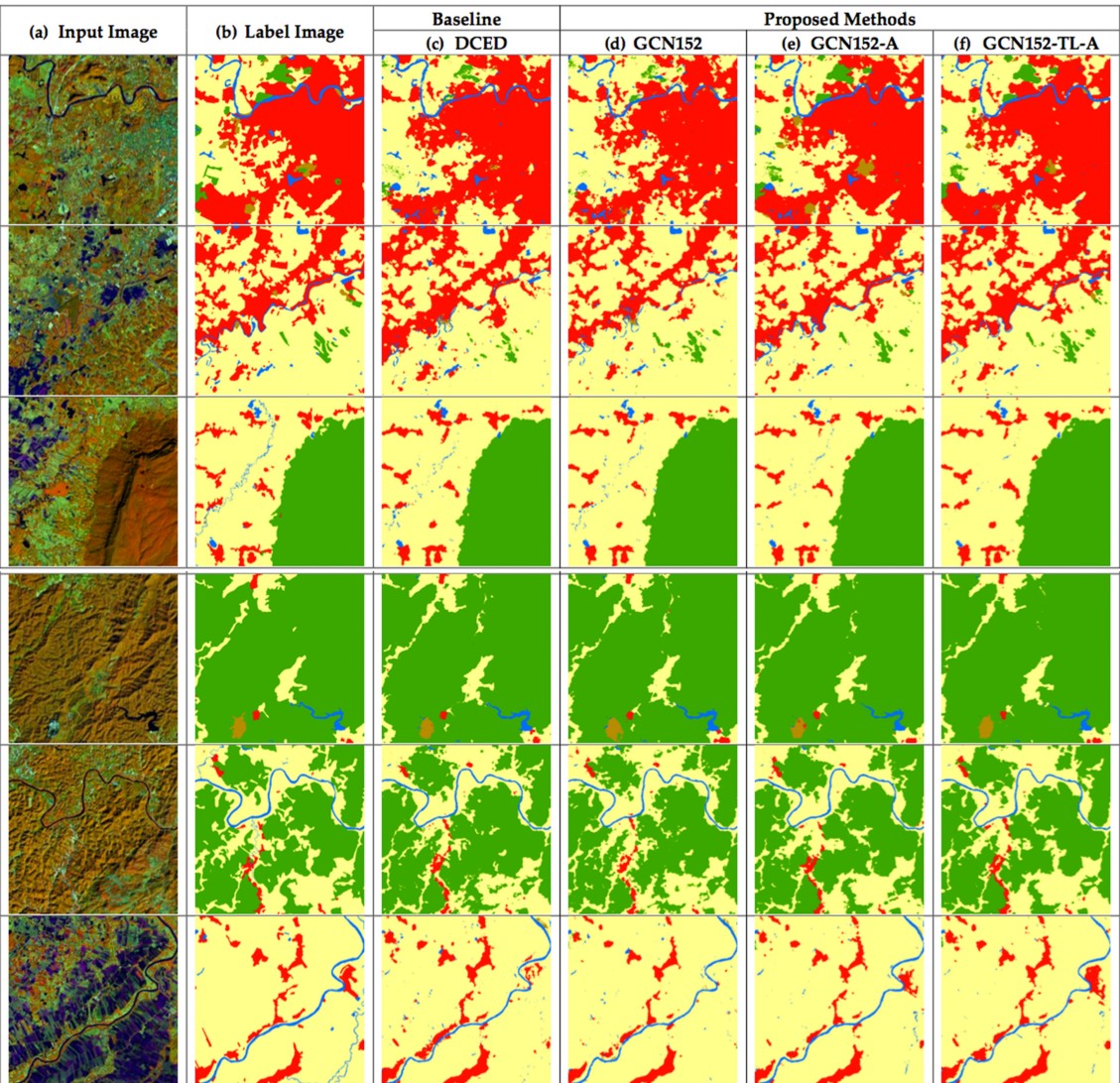

**Figure 9.** Six testing sample inputs and output satellite images on Landsat-8 in the Nan province in Thailand, where rows refer to different images. (**a**) Original input image. (**b**) Target map (ground truth). (**c**) Output of Encoder–Decoder (Baseline). (**d**) Output of GCN152. (**e**) Output of GCN152-A. and (**f**) Output of GCN152-TL-A. The label of medium resolution dataset includes five categories: Agriculture (yellow), Forest (green), Miscellaneous (Misc, brown), Urban (red) and Water (blue).

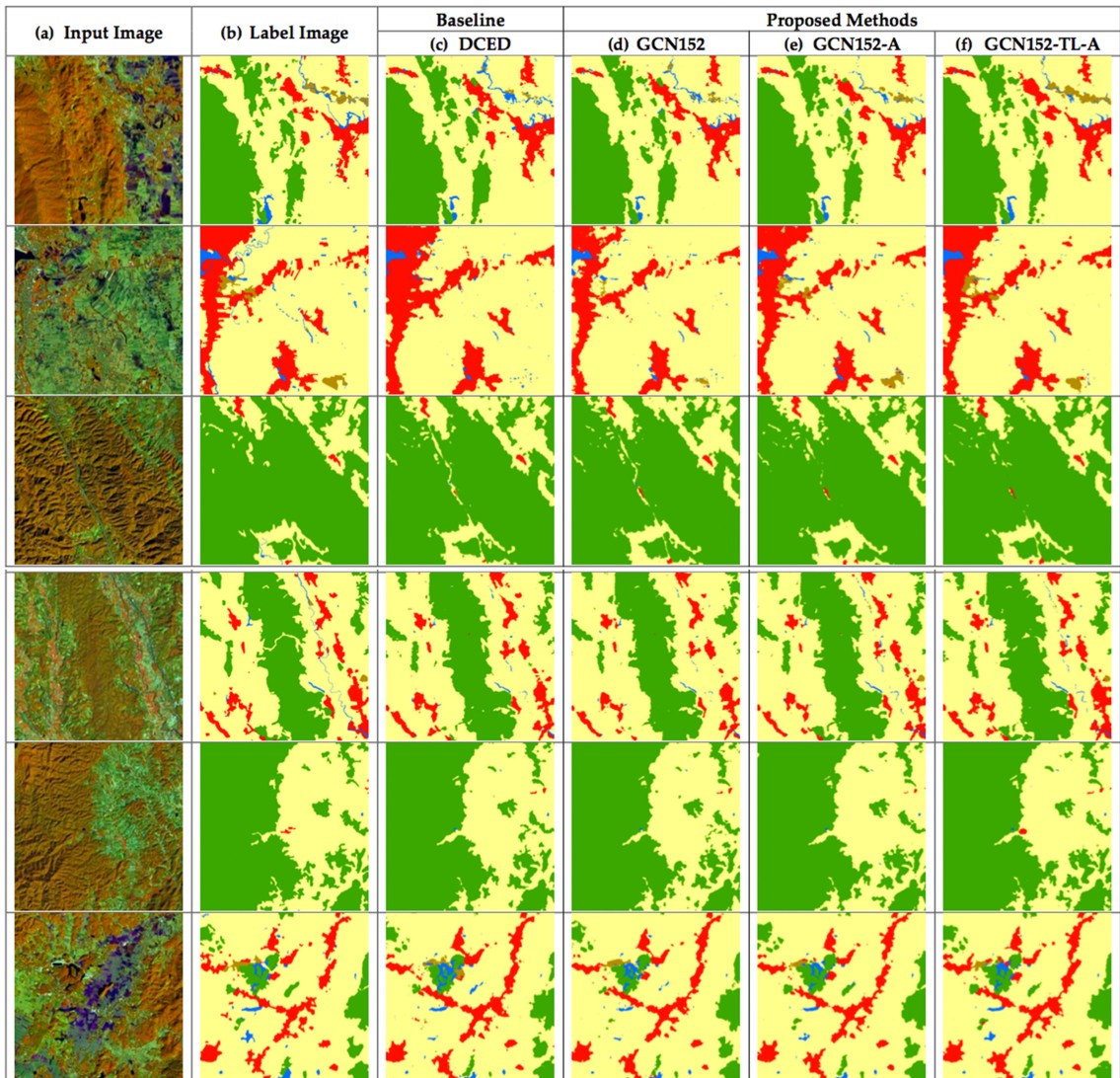

**Figure 10.** Six testing sample input and output satellite images on Landsat-8 in Nan in Thailand, where rows refer to different images. (**a**) Original input image. (**b**) Target map (ground truth). (**c**) Output of Encoder–Decoder (Baseline). (**d**) Output of GCN152. (**e**) Output of GCN152-A. and (**f**) Output of GCN152-TL-A. The label of medium resolution dataset includes five categories: Agriculture (yellow), Forest (green), Miscellaneous (Misc, brown), Urban (red) and Water (blue).

### 5.1.3. The Effect of Using Domain-Specific Transfer Learning on Landsat-8 Corpus

Our last strategy aims to use approach of domain-specific transfer learning (details in Section 3.3) by reusing the pre-trained weight from the GCN152-A model on the ISPRS Vaihingen corpus. From Tables 2 and 3, the *F*1 of the GCN152-TL-A method is the winner; it clearly outperforms not only the baseline but also all previous generations. Its *F*1 is higher than that of the DCED (baseline) at 17.80%. Its *mean IoU* is higher than that of the DCED at 17.94%. Additionally, the result illustrates that the concept of domain-specific transfer learning can enhance both precision (0.8293) and recall (0.8476).

Figures 9 and 10 show 12 sample results from the proposed method. By applying all strategies, the images in the last column (Figures 9f and 10f) are similar to the ground truths (Figures 9b and 10b). Furthermore, *F*1-results and *mean IoU* scores are improved for each strategy we added to the network as shown in Figures 9c–f and 10c–f.

To achieve the highest accuracy, the network must be configured and many epochs must be trained until all parameters in the network are converged. Figure 11a illustrates that the proposed network has been properly set and trained until it is converged and runs more smoothly than the

baseline in Figure 12a. Furthermore, Figures 11b and 12b show that a higher number of epochs tend to show a better *F*1 score. Thus, the number of chosen epochs based on the validation data is 49 (the best model for this dataset).

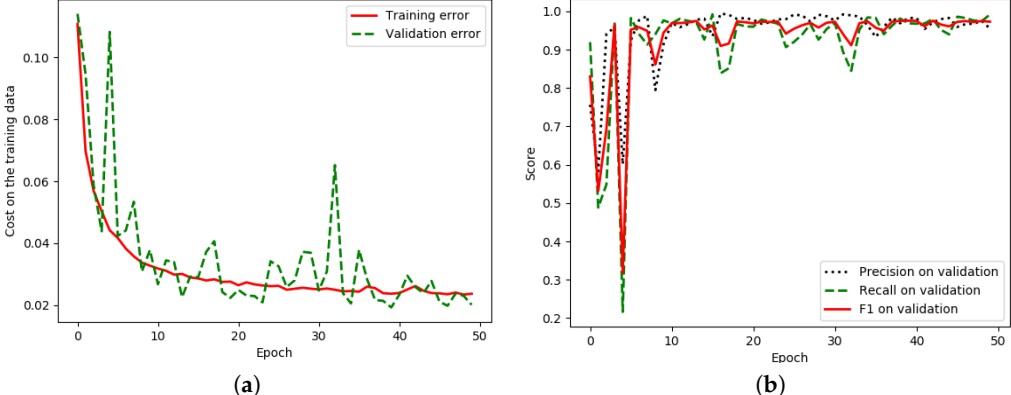

(a)                                                    (b)

**Figure 11.** Iteration plot on Landsat-8 corpus of the proposed technique, GCN152-TL-A; *x* refers to epochs and *y* refers to different measures (**a**) Plot of model loss (cross entropy) on training and validation datasets; (**b**) performance plot on the validation dataset.

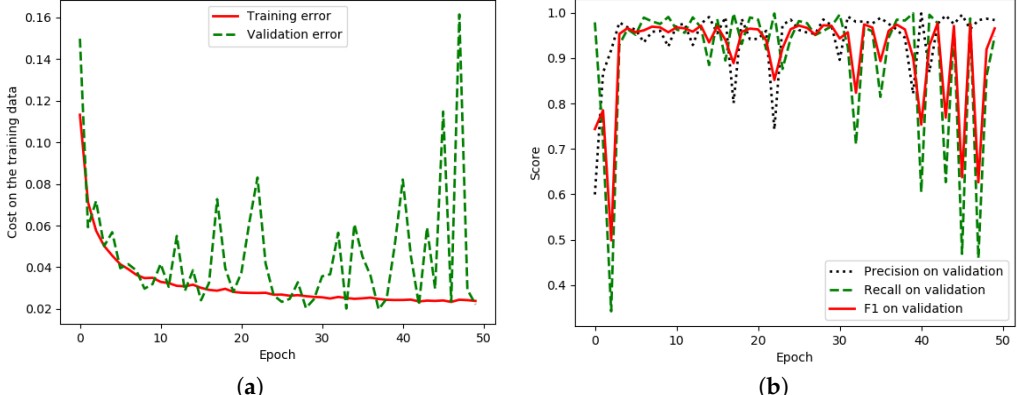

(a)                                                    (b)

**Figure 12.** Iteration plot on the Landsat-8 corpus of the baseline technique, the DCED [31–33]; *x* refers to epochs and *y* refers to different measures. (**a**) The plot of model loss (cross entropy) on training and validation datasets; (**b**) the performance plot on the validation dataset.

Twelve sample testing results (shown as Figures 9 and 10) are based on the proposed method with respect to Nan (one of the northern provinces (changwat) of Thailand and where agriculture is the main industry). The results of the last column look closest to the ground truth in the second column.

As can be seen in Figures 9 and 10, the performance of our best model outperforms other advanced models by a considerable margin on each category, especially for the agriculture, miscellaneous (Misc), and water classes. Furthermore, the loss curves shown in Figure 11a exhibit that our model performs better on all given categories.

## 5.2. Results of the ISPRS Vaihingen Challenge Corpus with Discussion

An experiment was conducted on the ISPRS Vaihingen Challenge corpus, and the result is shown in Tables 4 and 5 by comparing between baseline and variations of the proposed techniques. This shows that our network with all strategies (GCN152-TL-A) outperforms other methods. More details will be discussed to show that each of the proposed techniques can improve accuracy. Only in this experiment is there one baseline, which is the DCED network.

**Table 4.** Results of the testing data of the ISPRS 2D semantic labeling challenge corpus between the baseline and five variations of our proposed techniques in terms of *precision*, *recall*, *F*1, and *mean IoU*.

|  | Pretrained | Backbone | Model | *Precision* | *Recall* | *F1* | *mean IoU* |
|---|---|---|---|---|---|---|---|
| **Baseline** | - | - | DCED [31–33] | 0.7519 | 0.7925 | 0.7693 | 0.8651 |
| **Proposed Method** | - | Res50 | GCN [15] | 0.7636 | 0.7917 | 0.776 | 0.8776 |
|  | - | **Res101** | GCN | 0.7713 | **0.8059** | 0.7862 | 0.8972 |
|  | - | **Res152** | GCN | 0.7736 | 0.8021 | 0.7864 | 0.8977 |
|  | - | **Res152** | **GCN-A** | 0.7847 | 0.7961 | 0.7902 | 0.9057 |
|  | TL | **Res152** | **GCN-A** | **0.7888** | 0.8001 | **0.7942** | **0.9123** |

**Table 5.** Results of the testing data of ISPRS Vaihingen Challenge corpus between each class with our proposed techniques in terms of *Average Accuracy*.

|  | Model | IS | Buildings | LV | Tree | Car |
|---|---|---|---|---|---|---|
| **Baseline** | DCED [31–33] | 0.9590 | 0.9778 | 0.9108 | 0.9805 | 0.6832 |
| **Proposed Method** | GCN50 [15] | 0.9595 | 0.9628 | 0.9403 | 0.9896 | 0.7292 |
|  | GCN101 | 0.9652 | 0.9827 | **0.9615** | 0.9797 | 0.7387 |
|  | GCN152 | 0.9543 | **0.9962** | 0.9445 | 0.9754 | 0.7710 |
|  | **GCN152-A** | 0.9614 | 0.9865 | 0.9554 | 0.9871 | 0.8181 |
|  | **GCN152-TL-A** | **0.9664** | 0.9700 | 0.9499 | **0.9901** | **0.8567** |

### 5.2.1. Effect of the Enhanced GCN on the ISPRS Vaihingen Corpus

Our first strategy aims to increase the *F*1 and *mean IoU* score of the network by varying backbones using ResNet 50, ResNet 101, and ResNet 152 rather than the traditional one, the DCED method. From Tables 4 and 5, the *F*1 of GCN152 (0.7864) outperforms that of GCN50 (0.776), GCN101 (0.768), and the baseline method, DCED (0.7693); this yields a higher *F*1 at 0.02%, 0.68%, and 1.01%, respectively. The *mean IoU* of GCN152 (0.8977) outperforms that of GCN50 (0.8776), GCN101 (0.8972), and the baseline method, DCED (0.8651); this yields a higher *mean IoU* at 0.02%, 0.68%, and 1.01% respectively. This can imply that an enhanced GCN is also more accurate than the DCED approach on a very high resolution dataset. ResNet with a large number of layers is still more robust than a small number of layers, the same as that performed on the Landsat-8 corpus (Section 5.1.1).

When comparing the results between the original GCN method and the enhanced GCN methods on the Landsat-8 corpus (Table 4), it is clear that the GCN with a larger backbone layer can improve network performance in terms of *F*1 and *mean IoU*

### 5.2.2. Effect of Using Channel Attention on ISPRS Vaihingen Corpus

Our second mechanism focused on utilizing the channel attention block to change the weights of the features on each stage to enhance the consistency. From Tables 4 and 5, the *F*1 of GCN152-A (0.7902) is greater than that of GCN152 (0.7864); this yields a higher *F*1 score at 0.38%. The *mean IoU* of GCN152-A (0.9057) is better than that of GCN152 (0.8977); this yields a higher *mean IoU* score at 0.80%. The results (Figures 13e and 14e) show that this can also cause the network to obtain discriminative features stage-wise to make intra-class prediction consistent with respect to very high resolution images.

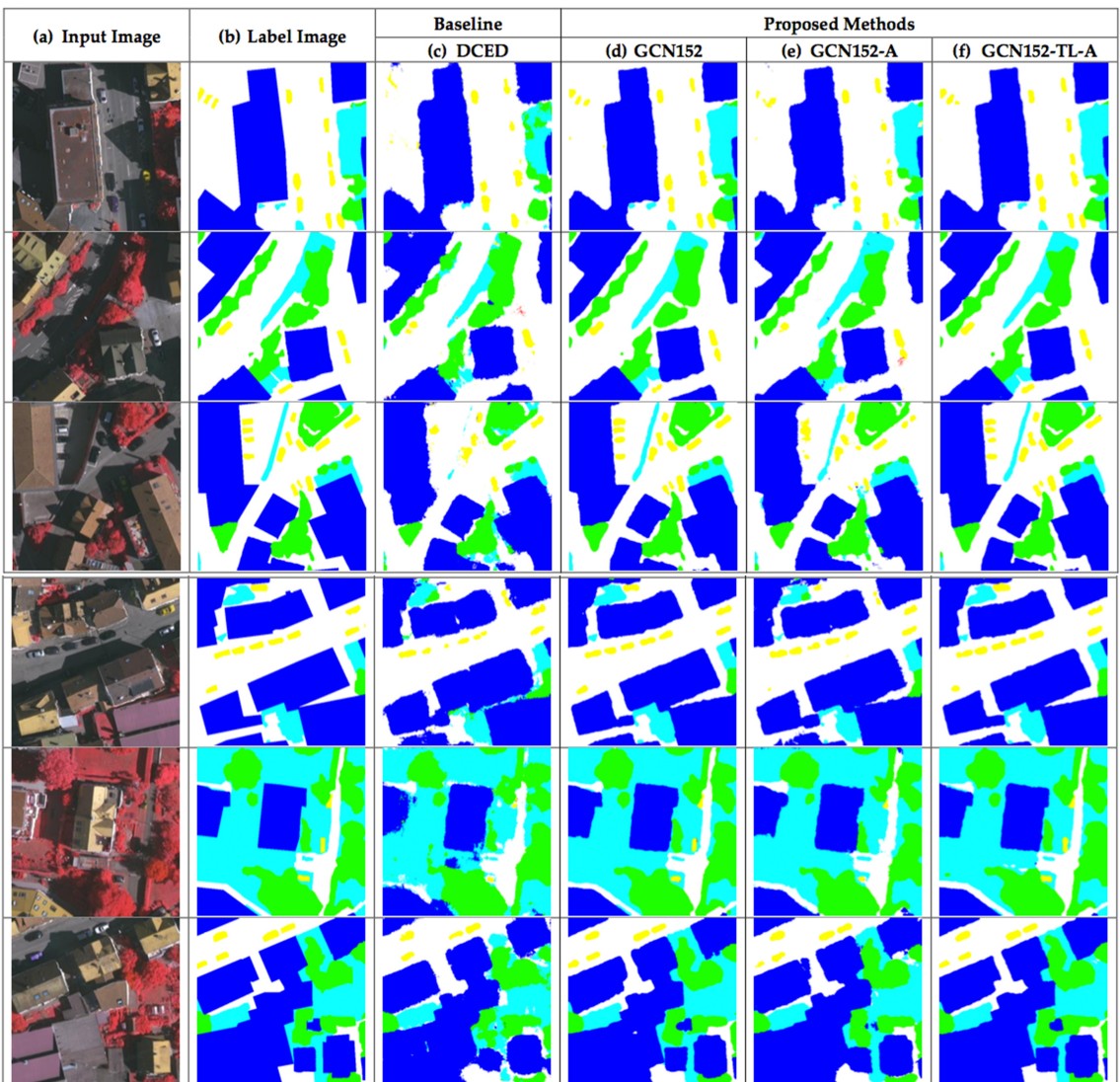

**Figure 13.** Six testing sample input and output aerial images on ISPRS Vaihingen Challenge corpus, where rows refer different images. (**a**) Original input image. (**b**) Target map (ground truth). (**c**) Output of Encoder–Decoder (Baseline). (**d**) Output of GCN152. (**e**) Output of GCN152-A. and (**f**) Output of GCN152-TL-A. The label of the Vaihingen Challenge includes six categories: impervious surface (imp surf, white), building (blue), low vegetation (low veg, cyan), tree (green), car (yellow) and clutter/background (red).

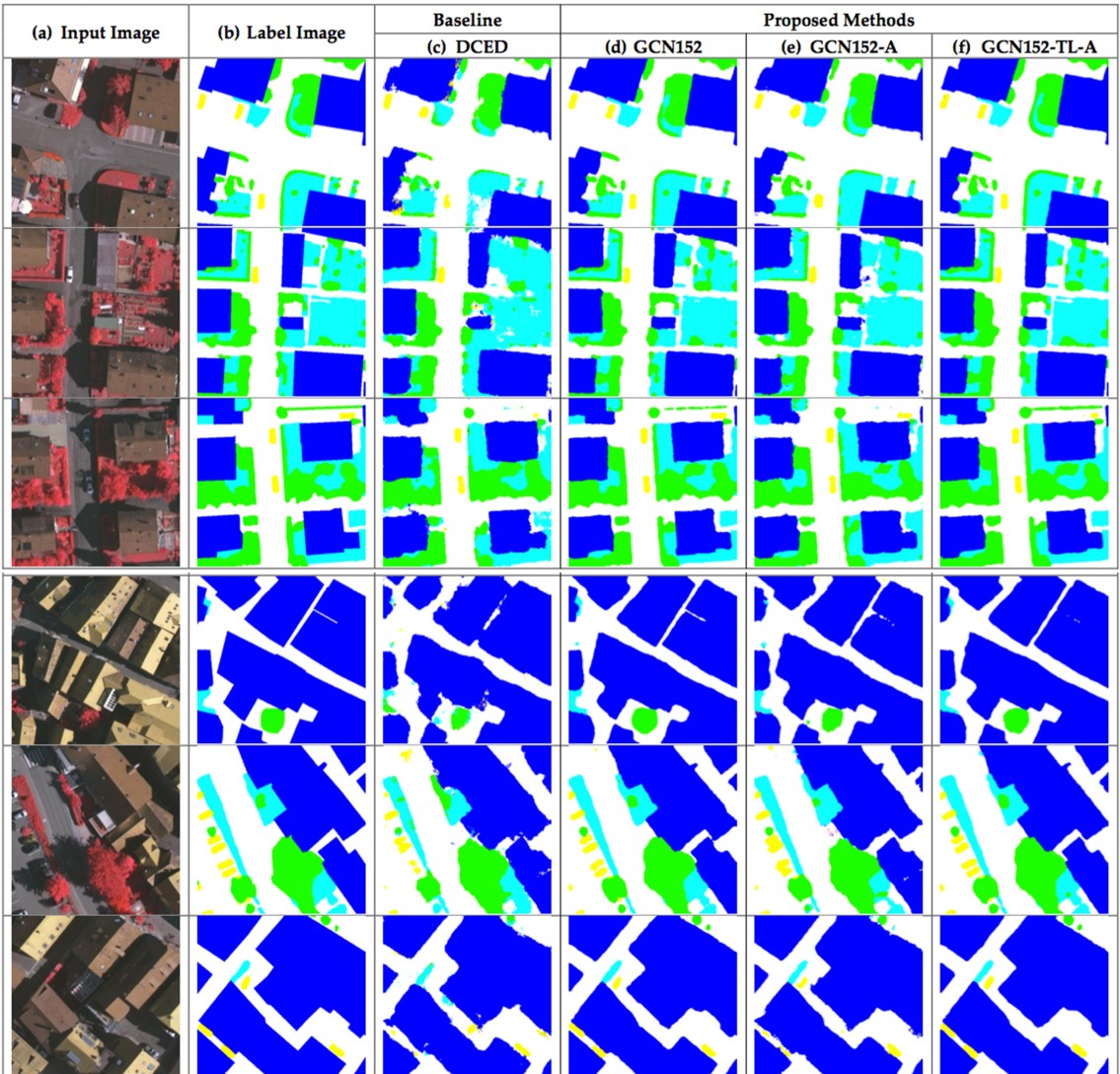

**Figure 14.** Six testing sample input and output aerial images on ISPRS Vaihingen Challenge corpus, where rows refer different images. (**a**) Original input image; (**b**) Target map (ground truth); (**c**) Output of Encoder–Decoder (Baseline); (**d**) Output of GCN152; (**e**) Output of GCN152-A; and (**f**) Output of GCN152-TL-A. The label of the Vaihingen Challenge includes six categories: impervious surface (imp surf, white), building (blue), low vegetation (low veg, cyan), tree (green), car (yellow), and clutter/background (red).

5.2.3. The Effect of Using Domain-Specific Transfer Learning on the ISPRS Vaihingen Corpus

Our last strategy aims to perform domain-specific transfer learning (details in Section 3.3) by reusing the pre-trained weight from the GCN152-A model on the Landsat-8 corpus. From Tables 4 and 5, the *F*1 of the GCN152-TL-A method is the winner; it clearly outperforms not only the baseline but also all previous generations. Its *F*1 is higher than the DCED (baseline) at 2.49% and 1.82% consecutively. Its *mean IoU* is higher than the DCED and the GCN at 4.76% and 3.51%, respectively. Additionally, the result illustrates that the concept of domain-specific transfer learning can enhance both precision (0.7888) and recall (0.8001).

Figures 13 and 14 shows 12 sample results from the proposed method. By applying all strategies, the images in the last column (Figures 13f and 14f) are similar to ground truths (Figures 13b and 14b). Furthermore, *F*1 results and *mean IoU* scores are improved for each strategy we added to the network as shown in Figures 13c–f and 14c–f.

To further evaluate the effectiveness of the proposed GCN152-TL-A comparisons with the baseline method on the one challenging benchmark and the one private benchmark are presented in Tables 2 and 3 for the Landsat-8 dataset with respect to Nan (Thailand) and Tables 4 and 5 for the Vaihengen dataset. All extensive experiments on the Landsat-8 and ISPRS datasets demonstrate that the proposed method clearly achieves promising gains compared with the baseline approach.

Figures 13 and 14 show twelve sample testing results from the proposed method on ISPRS Vaihingen corpus. The results of the last column are also similar to the ground truth in the second column same as performed on Landsat-8 corpus. Considering to each class (are shown in Tables 3 and 5), almost every classes (three out of five) from our proposed methods are the winner in term *Average Accuracy*.

As can be seen in Figures 13 and 14, the performance of our best model outperforms other advanced models by a considerable margin on each category, especially for the impervious surface (IS), tree, and car categories. To show the effectiveness of the proposed methods, we performed comparisons against a number of state-of-the-art semantic segmentation methods, as listed in Table 4, Table 5 with respect to the ISPRS corpus, and Tables 2 and 3 with respect to the Landsat-8 corpus. The DCED [31–33] and GCN [15] are the versions with ResNet-50 as their backbone. In particular, we re-implemented the DCED with Tensorflow-Slim [36], since the released code was built on Caffe [38]. We can see that our proposed methods significantly outperform other methods on both the *F*1 *score* and *mean IoU*.

In terms of the computational cost, our framework requires slightly additional training time compared to the baseline approach, DCED, by about 6.25% (6–7 h), and GCN, by about 4.5% (4–5 h). In our experiment, DCED's training procedure took approximately 16 h per dataset, and finished after 50 epochs with 1152 s per epoch. Our framework is a modification of the GCN-based deep learning architecture. The channel attention model increases the time by 20 min compared with the GCN152 method. There is no additional time required when reusing pre-trained weights.

## 6. Conclusions and Future Work

In this study, we propose a novel CNN framework to perform semantic labeling on remotely sensed images. Our proposed method achieves excellent performance by presenting three aspects. First, a global convolutional network (GCN) is employed and enhanced by adding larger numbers of layers to better capture complex features. Second, channel attention is proposed to assign a proper weight for each extracted feature on different stages of the network. Finally, domain-specific transfer learning is introduced to allay the scarcity issue by training the initial weights using other remotely sensed corpora whose resolutions can be different. The experiments were conducted on two datasets: Landsat-8 (medium resolution) and the ISPRS Vaihingen Challenge (very hign resolution) datasets. The results show that our model that combines all proposed strategies outperforms baseline models in terms of *F*1 and mean IoU. The final results show that our enhanced GCN outperforms the baseline (DCED)—17.48% for F1 on the Landsat-8 corpus and 2.48% on the ISPRS corpus.

In the future, more choices of semantic labeling, modern optimization techniques, and/or other novel activation functions will be investigated and compared to obtain the best GCN-based framework for semantic segmentation in remotely sensed images. Moreover, incorporating other data sources (e.g., a digital surface model) might be needed to increase the accuracy of deep learning for both the CNN and the modern deep learning layer with very low confidence simultaneously. These aforementioned issues will be investigated in future research.

**Author Contributions:** T.P. performed all the experiments and wrote the paper; P.V. and T.P. performed the results analysis and edited the manuscript. K.J., S.L., T.P. and P.S. reviewed the results. T.P. revised the manuscript.

**Funding:** This research received no external funding.

**Acknowledgments:** T. Panboonyuen thanks the scholarship from The 100th Anniversary Chulalongkorn University Fund granted and The 90th Anniversary Chulalongkorn University Fund (Ratchadaphiseksomphot Endowment Fund). We greatly acknowledge Geo-informatics and Space Technology Development Agency (GISTDA), Thailand, for providing satellite imagery used in this study and T. Panboonyuen thanks to the staff from the GISTDA (Thanwarat Anan, Suwalak Nakya, Bussakon Satta) for the supply of LANDSAT-8 imagery and supporting ground data.

**Conflicts of Interest:** The authors declare no conflict of interest.

## Abbreviations

The following abbreviations are used in this manuscript:

| | |
|---|---|
| BR | Boundary Refinement |
| CNN | Convolutional Neural Network |
| DCED | Deep Convolutional Encoder–Decoder |
| GCN | Global Convolutional Network |
| MR | Medium Resolution |
| RGB | Red–Green–Blue |
| LS | Landsat |
| TL | Transfer Learning |
| VHR | Very High Resolution |

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
