# Peer review of "Semantic Segmentation on Remotely Sensed Images Using an Enhanced Global Convolutional Network with Channel Attention and Domain Specific Transfer Learning"

_remotesensing, doi:10.3390/rs11010083_

Round 1
Reviewer 1 Report
[row 5, 7] "CNN network": if abbreviation is opened, the result is "convolutional neural network network".
[row 126, 131, 132 and 145]: it is good style to consider the sequence of references into figures. The firstly mentioned figure is showed first and its location must be closer to the first mention.
[row 1] "to automatically annotate semantics": according to the subject of research, the semantic classification is completed. Semantics, in this study case, are classes added to pixels (water, vegetation etc.). So, "to annotate semantics" will be addition of description to each class. Phrase isn't logical.
Author Response
Point 1:
[row 5, 7] "CNN network": if abbreviation is opened, the result is "convolutional neural network network".
Response 1:
We have already revised “CNN network” as “CNN”.
Point 2:
[row 126, 131, 132 and 145]: it is good style to consider the sequence of references into figures. The firstly mentioned figure is showed first and its location must be closer to the first mention.
Response 2:
We have revised to consider the sequence of references into figures. However, we try to arrange the firstly mentioned figure and we see its location is nearest to the first mention.
Point 3:
[row 1] "to automatically annotate semantics": according to the subject of research, the semantic classification is completed. Semantics, in this study case, are classes added to pixels (water, vegetation etc.). So, "to annotate semantics" will be addition of description to each class. Phrase isn't logical.
Response 3:
We agree that and have revised "to automatically annotate semantics” as "to annotate semantics".

Reviewer 2 Report
This paper presents a DCED network with enhanced global convolutional Network with channel attention and domain specific transfer learning to obtain better GCN feature extraction and classification for remotely-sensed images. It is quite meaningful for deep learning efficiency improvement because it provides another possibility for feature extraction and integration. Research design can effectively improve the performance that previous studies can't achieve, and the contents in every paragraph and comparison process seem to be able to generate persuasive instructions. However, some possibility and necessity measures such as random horizontal flip, sampling strategies, and suitable map scale choosing should be wrote in clear, easy-to-understand memorandum to make sure the research can be verified by others. It is very clear in the narrative part of the structure of the article and the research methods, but whether some detailed part of testing objectives and workflows (e.g., terrain complexity from DSM & NDSM, assignment of weights of the remote sensing features) in the research may have higher uncertainty. For this reason, research restrictions and limitations should be stated. Basically, the professionalism and creativity of this article is worthy of recognition. It would be a good article for publishing after minor revision. Here with concerns need to be addressed:
Question & Comment1:
The previous research part was written very completely, and you have already presented the constructive revision to enhance interest feature and weak background interference with your methods. I think it would be interesting to explain in more detail about the assignment of weights of “channel attention block & visual attention mechanism” or to add some comparision information in discussion paragraph. It would be quite helpful for readers to quickly understand the contribution of your research.
Question & Comment2
In this study, whether the different initial window sizes for feature extraction, and the arrangement of convolution layers, pooling layers, and transfer learning affect the research results have not been carefully explored. GCN itself has certain gray box characteristics, so it is necessary to mention the uncertainty of the research.
Question & Comment3:
I am not sure whether or not your method can directly applied other remotely-sensed images or UAV images without correction and cropping processes and show promising results, because it is very common to use unoptimized imgaes directly in fast accessment of land-use & land cover data production. I suggest the applicable materials (e.g., image quality, frame size, map scale and spatial resolution) can be defined more detailed.
Question & Comment4:
In your research, you metioned that the attention mechanism with weight mask could enhance the interest features and weakens the irrelevant feature in the CNN feature map. I am very curious whether the " variations of backbones and channel attention block" will be affected by the “density of objects”, “the overlapping of objects”, and “the image distortion of objects”?
Question & Comment5:
Feature convolution and pooling can solve the problem of knowledge adaptation from multiple scene datasets, and reduce the network parameters and the spatiotemporal size of the image representations. However, the image distortion and shadow effects seemed to be an unignorable problem and have great influence in frature recognition and image classification. If the input of the training sample from original images is highly discriminating (e.g., light changes, shape changes, image distortion) is it possible to correct this problem automatically through the research process?
Author Response
Point 1:
The previous research part was written very completely, and you have already presented the constructive revision to enhance interest feature and weak background interference with your methods. I think it would be interesting to explain in more detail about the assignment of weights of “channel attention block & visual attention mechanism” or to add some comparision information in discussion paragraph. It would be quite helpful for readers to quickly understand the contribution of your research.
Response 1:
Because we used “channel attention block” citation from [16,17] so we do not explain in more detail about the assignment of weights. However, we have already added more detail about the assignment of weights in Section 3.3 (explains via math formulas in more detail about the assignment of weights)
Point 2:
In this study, whether the different initial window sizes for feature extraction, and the arrangement of convolution layers, pooling layers, and transfer learning affect the research results have not been carefully explored. GCN itself has certain gray box characteristics, so it is necessary to mention the uncertainty of the research.
Response 2:
For this study, we have used the default of initial window sizes for feature extraction, and the arrangement of convolution layers, pooling layers, and transfer learning from the original papers (both DCED and GCN). Therefore, this work proposed a novel CNN framework to perform semantic labeling on remote-sensed images.
Point 3:
I am not sure whether or not your method can directly applied other remotely-sensed images or UAV images without correction and cropping processes and show promising results, because it is very common to use unoptimized imgaes directly in fast accessment of land-use & land cover data production. I suggest the applicable materials (e.g., image quality, frame size, map scale and spatial resolution) can be defined more detailed.
Response 3:
We have already shown the applicable materials (e.g., image quality, frame size, map scale and spatial resolution) in Section 3.1, 4.1 and 4.2.
Point 4:
In your research, you metioned that the attention mechanism with weight mask could enhance the interest features and weakens the irrelevant feature in the CNN feature map. I am very curious whether the " variations of backbones and channel attention block" will be affected by the “density of objects”, “the overlapping of objects”, and “the image distortion of objects”?
Response 4:
In our research, refer to our experiments, the results of medium-resolution and very high-resolution testing image (by using our proposed modern CNN method) was clearly shown variations of backbones and channel attention block will be not affect by the “density of objects”, “the overlapping of objects”, and “the image distortion of objects”.
Point 5:
Feature convolution and pooling can solve the problem of knowledge adaptation from multiple scene datasets, and reduce the network parameters and the spatiotemporal size of the image representations. However, the image distortion and shadow effects seemed to be an unignorable problem and have great influence in frature recognition and image classification. If the input of the training sample from original images is highly discriminating (e.g., light changes, shape changes, image distortion) is it possible to correct this problem automatically through the research process?
Response 5:
We pretty sure to correct this problem automatically through our research process. However, our future research will be proposed robust algorithm anew with our proposed modern CNN method on remote sensing images.

Reviewer 3 Report
The authors proposed a method of deep-learning-based segmentation of satellite images with combinations of GCN, attention channel, and transfer learning between medium- and high-resolution satellite images. The proposed method is innovative for the practical needs of the rapidly growing satellite data resources and potential social benefits. Therefore, I would like to recommend this for publication in the Remote Sensing though I would request authors to address my comments below before publication.
1. Section 3.4 - Please add a little more description for beginners of transfer learning, specifically mention what the algorithms transfer. Citation from [18] and the original references under the section "Combining Transfer Learning and Deep Learning" in [18] would be useful, such as "we can transfer low-level features, such as edges and corners, and learn new high-level features specific to the target problem."
2. Some more discussions on the accuracy by category are preferred. For example in Figure 9 and Table 3, accuracy on urban was degraded by the transfer learning, which likely induces misclassification of agriculture to urban (just from my observation to Figure 9 and Table 3; it should be based on error matrixes). It would be due to texture similarity between agriculture and urban which were better to be discriminated by spectral differences while the transferred model from high-resolution images likely weighted texture more. Such kind of discussion will be useful in applications of the method to the other data and areas.
3. Following to the comment above #2, please consider adding error matrixes as an appendix or supplemental materials.
4. Figure 1--Please indicate the abbreviation "BR" in the caption.
5. Figure 4--It looks something missing in the bottom in the middle of the arrow line between "Concatenation" and "Sum."
6. Figure 9, 12, 13, 14--Please add a color legend in the caption as well as Figure 6 and 8. It will be repeating though useful for readers not to come back to Figure 6 and 8.
Author Response
Point 1:
Section 3.4 - Please add a little more description for beginners of transfer learning, specifically mention what the algorithms transfer. Citation from [18] and the original references under the section "Combining Transfer Learning and Deep Learning" in [18] would be useful, such as "we can transfer low-level features, such as edges and corners, and learn new high-level features specific to the target problem."
Response 1:
Section 3.4 has been revised. Also, we have been added a little more description for beginners of transfer learning, specifically mention what the algorithms transfer and refers to citation from [18] and the original references under the section "Combining Transfer Learning and Deep Learning" in [18] to represent the useful sentence such as "we can transfer low-level features, such as edges and corners, and learn new high-level features specific to the target problem.".
In the case of problems in the computer vision and remote sensing domain, certain low-level features, such as edges, shapes, corners and intensity, can be shared across tasks, and thus enable knowledge transfer among tasks. Also, knowledge from an existing task acts as an additional input when learning a new target task.
Point 2:
Some more discussions on the accuracy by category are preferred. For example in Figure 9 and Table 3, accuracy on urban was degraded by the transfer learning, which likely induces misclassification of agriculture to urban (just from my observation to Figure 9 and Table 3; it should be based on error matrixes). It would be due to texture similarity between agriculture and urban which were better to be discriminated by spectral differences while the transferred model from high-resolution images likely weighted texture more. Such kind of discussion will be useful in applications of the method to the other data and areas.
Response 2:
A discussion is provided in Section 5.1 and 5.2. Also, we agree that if our work shows error matrices. However, even though accuracy on urban and forest were degraded by the transfer learning, which likely induces misclassification of agriculture to urban but the latest of our proposed method show almost every classes (three out of five) from our proposed methods are the winner so we think our discussion will be useful in applications of the method to the other data and areas.
Point 3:
Following to the comment above #2, please consider adding error matrixes as an appendix or supplemental materials.
Response 3:
We have added error matrixes as supplemental materials (both results on ISPRS Vaihingen Challenge Corpus and ISPRS corpus).
Point 4:
Figure 1--Please indicate the abbreviation "BR" in the caption.
Response 4:
We have already indicated the abbreviation "BR" in the caption at Figure 1.
Point 5:
Figure 4--It looks something missing in the bottom in the middle of the arrow line between "Concatenation" and "Sum."
Response 5:
We have revised the components of Channel Attention Block. From our work, we have just used the red line as the downsample operators.
Point 6:
Figure 9, 12, 13, 14--Please add a color legend in the caption as well as Figure 6 and 8. It will be repeating though useful for readers not to come back to Figure 6 and 8.
Response 6:
We have already added a color legend in the caption at Figure 9, 12, 13, 14.

Round 2
Reviewer 1 Report
There is the misunderstand with "to annotate semantics", it wasn't phrase for replacement - this combination isn't logical in your content. "To annotate semantics" is usable, but it belongs to domain "semantic web", "ontologies", "knowledge databases", etc. Of course, readers will understand what you mean, however, it isn't correct definition for your completed task (classification). Therefore, standard definitions like "classification", "detection", "recognition" are usable in your research domain. If you want to input word "semantic", there is definition "semantic segmentation". This phrase can be replaced by:
* ", it is crucial to complete land cover classification on the raster images";
* ", it is crucial to complete semantic segmentation on the raster images";
* ", it is crucial to detect and to recognize, e.g.,river,building,forest,etc, on the raster images";
* etc.
Reviewer 2 Report
All the comments had reply with acceptable correction.
Reviewer 3 Report
I confirmed the revision addressed my comments.